# The First Report of the Genera *Abaria* and *Drepanocentron* (Trichoptera: Xiphocentronidae) from China, with Descriptions of Two New Species

**DOI:** 10.3390/insects13010095

**Published:** 2022-01-14

**Authors:** Lang Peng, Xinyu Ge, ChangHai Sun, Beixin Wang

**Affiliations:** Insect Classification and Aquatic Insect Laboratory, College of Plant Protection, Nanjing Agricultural University, Nanjing 210095, China; 2020202045@stu.njau.edu.cn (L.P.); 2019202038@njau.edu.cn (X.G.)

**Keywords:** newly recorded, caddisfly, Xiphocentronidae, morphology

## Abstract

**Simple Summary:**

The small family Xiphocentronidae Ross 1949 (186 species and 7 genera) has a wide distribution and high diversity in the Oriental region. Despite the rich biodiversity of Trichoptera in eastern China, only one xiphocentronid genus (*Melanotrichia*), with three species, is recognized. The status of Xiphocentronidae as an evolutionary lineage distinct from Psychomyiidae has been subject to debate in the past. The specimens in this study provide additional records on the distribution of two genera in China, which are conducive to biological and morphological studies and can also provide molecular data support for further studies.

**Abstract:**

The genera *Abaria* Mosely 1948 and *Drepanocentron* Schmid 1982 are recorded in China for the first time. In this study, two new species, *Abaria herringbona* sp. nov., from Guang-xi, and *Drepanocentron fuxiensis* sp. nov., from An-hui, are described and illustrated. Male genitalia of these two new species is distinguishable from those of other *Abaria* and *Drepanocentron* species. In addition, *Melanotrichia attia* Malicky & Chantaramongkol 1992 is a new record for the Chinese caddisfly fauna.

## 1. Introduction

The serpentine-tube-dwelling [1,2] caddisfly family Xiphocentronidae was established by Ross [3] for a unique small black caddisfly from Mexico. When Edwards [4] described the larva of *Xiphocentron*, he recognized that the larva was similar to the larva of Psychomyiidae (as currently defined), and he proposed that Xiphocentronidae should be synonymous with it [5]. Schmid [6] revealed the considerable diversity and wide distribution of this family in a monograph devoted to it. Since then, the valid status of the Xiphocentronidae has been generally recognized.

As currently defined, the family Xiphocentronidae is a small family of seven genera with 186 species [7,8,9], including *Abaria* Mosely (39 species, from the Oriental, Afrotropical, and Western Palearctic Regions), *Cnodocentron* Schmid (13 species, from the southwestern United States of America, northern South America, India, and Southeast Asia), *Drepanocentron* Schmid (42 species, all from the Oriental Region), *Machairocentron* Schmid (6 species, from Central America and northern South America), *Melanotrichia* Ulmer (30 species, from the eastern Palearctic and Oriental regions), *Proxiphocentron* Schmid (5 species, from Southeast Asia), and *Xiphocentron* Brauer (51 species, widespread in the Neotropical region, extending into Mexico, the southwestern USA, and the Greater Antilles) [5]. Most of them are distributed in the Oriental region (118 species) and the Neotropical region (59 species). The only described Xiphocentronidae fossil, belonging to the genus *Xiphocentron*, was found in Miocene Mexican amber [10].

However, the family is poorly known in China. Thus far, only one genus, i.e., *Melanotrichia*, and 3 of its 30 species (*Melanotrichia acclivopennis* (Hwang, 1957); *Melanotrichia hwangi* (Ross, 1949); *Melanotrichia serica* Barnard & Dudgeon, 1984) are recognized in China [8,11], although eastern China is known for great biodiversity of Trichoptera. In the present study, two genera of Xiphocentronidae, *Abaria* and *Drepanocentron*, are recognized in China for the first time, and one new species of each genus is described. In addition, *Melanotrichia attia* Malicky & Chantaramongkol 1992, collected in Gui-zhou, is a new record for the Chinese caddisfly fauna.

The genus *Abaria* was erected by Mosely [12] with *Abaria tripunctata* Mosely 1948 as its type species. Of the known 39 *Abaria* species [8], 37 species are endemic to the Oriental region, among which 15 were described from Vietnam by Oláh, Johanson, and Malicky [12,13], 9 from India by Schmid and Oláh [6,14], 7 from Thailand by Malicky and Chantaramongkol [15,16,17], 4 from the Philippines by Mey [18,19], 1 from Indonesia by Malicky, Melnitsky and Ivanov [20], 1 from Sri Lanka by Schmid [21], 1 by Marlier from Afrotropical Congo [22], and 1 by Mosely from Yemen in both the Afrotropical and Western Palearctic Regions [12].

Based on the preanal appendages, Schmid [6] assigned his nine new species and two recognized species of *Abaria* into two species groups, i.e., the *A. tripunctata* species group and the *A. madhavi* species group. Later, Malicky, and Chantaramongkol [16] assigned two Thai species to the *A. madhavi* species group and another one to the *A. tripunctata* species group, but they also found that some species were characterized by preanal appendages unlike those of any of Schmid’s species groups [15]. Mey described two Philippine species of the *A. tripunctata* species group [18] and, of his other two Philippines species, one may belong to the *A. tripunctata* species group, but the other one remains unsettled. Later, Oláh and Johanson [13] applied Schmid’s species group system to their study of Vietnamese Xiphocentronidae, assigning 11 new species to the *A. tripunctata* species group and the other three new species to the *A. madhavi* species group.

The genus *Drepanocentron* was established by Schmid in 1982, with *Drepanocentron druhyu* as type species. Based on the forewing venation, structures of the inferior appendages, and of the aedeagus, Schmid [17] divided his 17 Indian new species into 2 species groups, i.e., *druhyu* and *citrangoda* species groups. Later, 24 species of the genus were reported by various authors, among them, 1 from Indonesia by Oláh and Malicky [23], 4 from Malaysia by Oláh [24] and Malicky [15,25], 5 from the Philippines by Mey [18,19,26], 3 from Thailand by Malicky and Chantaramongkol [13,15], and 12 from Vietnam by Malicky [15], Oláh, and Johanson [13] and Genco et al. [9]. Thus far, all 42 species of the genus *Drepanocentron* are exclusively distributed in the Oriental region.

This study brings the total number of known Chinese genera to 3 [11], the total numbers of species of genus *Abaria* to 40, and genus *Drepanocentron* to 43.

## 2. Materials and Methods

The materials were collected in 2015–2021 at three locations. Of all 9 individuals, 5 from Guang-xi collected by Malaise trap [27] were identified as *Abaria herringbona* n. sp., 1 from An-hui by light trap [28] as *Drepanocentron fuxiensis* n. sp., 2 from Gui-zhou by Malaise trap as *Melanotrichia attia* Malicky & Chantaramongkol 1992, and 1 from Zhejiang by light trap as *Melanotrichia acclivopennis* (Hwang, 1957).

All specimens were stored in 95% ethyl alcohol immediately after collection. The methods used for the preparation of specimens followed the methods of Xu [29]. For genitalia preparation, the male abdomen was cut from the body as close to the basal region of the abdomen as possible. The separated abdomen with its terminal genitalia was cleared using a 10% solution of sodium hydroxide (NaOH) at 80 °C temperature for about 20 min, to remove all the non-chitinous tissues and transferred into distilled water to rinse off the remaining NaOH. Then, the translucent abdomen was placed on a depression slide with 85% ethyl alcohol for examination [30]. Genitalia structure and wings of males were traced in pencil with a Nikon Eclipse 80i microscope equipped with a camera lucida. Pencil drawings were scanned with scanner Epson Perfection V30 SE and then placed as templates in Adobe Photoshop v.8.1 software program and inked digitally with a Wacom Intuos tablet and pen (CTL-671/KO-F) to produce illustrations. For each species, illustrations of male genitalia in lateral, dorsal, and ventral views and wing veins were prepared. Then, these body structures were stored in a microvial together with the remainder of the specimen in 95% ethanol.

The terminology for wing venation and male genitalia follows Schmid [6], as indicated in the figures and text. The terminology for setal warts is from Oláh and Johanson [31].

All types are deposited in the Insect Collection, Nanjing Agricultural University, Nanjing, Jiangsu Province, P.R. China (NJAU).

## 3. Results

### 3.1. Taxonomy

Family Xiphocentronidae Ross 1948.

All species described below have the following characters, except when otherwise stated:

Head: With a pair of enlarged frontal lateral compact setose warts, occupying most of the frontal area in front view, closer at the top than at the bottom. Head dorsum has four pairs of compact setose warts: small postgenal compact setose warts; large occipital compact setose warts; vertexal ocellar compact setose warts; vertexal lateroantennal compacts setose warts. Single frontal interantennal compact setose wart shifted dorsad between scapes and coronal groove, delineated by frontal grooves. Compact setal warts are absent on the cervix and cervical sclerites. Maxillary palp formula: I–II–III–IV–V, gradually increasing in length from segment I to segment IV, and segment V is longer than each of segments I–IV and much shorter than segments I–IV together. Thorax: Mesoprescutum is narrowing. Spurs 0–3, 4, 3–4, hind tibiae of male with apical spur simple or modified. Venation complete or reduced, forewings are with all forks present, or with only F2, or F2 and F4, or Fl, F2, and F4 present. Hind wings are either with no fork present or with F2 and F5 present [13,31].

Female genitalia: Elongate, with long apodemes on segments VIII, IX, and X; apicolateral papillae are absent [6].

Male genitalia: Segment VIII is with tergum and sternum fused or separated; tergum IX is reduced or developed; segment X is somewhat roof shaped. Preanal appendages are elongated and robust; intermediate appendages can be present or absent; inferior appendages are usually two segmented, or with two articles fused, and first articles are sometimes fused basally to each other, some with spines on the inner margin. Phallic apparatus is narrow and elongated, reaching abdominal segment V internally, and not articulated with any structure of genitalia. The phallic apparatus is enclosed by tergum X [32].

### 3.1.1. Genus Abaria Mosely 1948

Type species. *Abaria tripunctata* ME Mosely 1948.

Description: Specimen is blackish, and its mid- and hind legs are bicolored, brown, and yellow. Spurs are usually ♂ 0-4-3, ♀ 0-4-4; but ♂ 1-4-3; ♀ 1-4-4 in *Abaria tripunctata*. Forewings are dark brown, with several white spots. Forewings each with fork II are present, Sc simple, M bifurcate, Cu_1_ ending at wing margin, Cu_2_ weak, with two anal veins. Hind wings each with Sc absent, R_1_ distinct, Rs and M bifurcate, stems of R and M separated widely, producing large cells in the center of hind wing, with one short anal vein.

Male genitalia: Segment VIII tergite anterior and apical margins are indented or produced. Segment IX is with tergum membranous or entirely missing, sternite robust, with anterior margins having pair of slender apodemes. Preanal appendages are slender or enlarged. Segment X is triangular in lateral view. Inferior appendages are with basal and distal segments fused, usually strongly sclerotized, slender. Aedeagus is extremely long.

The species groups according to Schmid (1982) are defined by the following characteristics:

*Abaria tripunctata* species group: Preanal appendages are slender, cylindrical, sometimes each constricted at half length;

*Abaria madhqvi* species group: Preanal appendages are enlarged, inferior border sinuous, forming a prominent angle.

### 3.1.2. Genus Drepanocentron Schmid 1982

Type species. *Drepanocentron druhyu* F Schmid 1982.

Description: Brown species, thorax covered with black scales. Midlegs are bicolored, brown, and yellow. Forewings are often with black scale-like hairs. Hind wings bristled or with white, ivory, red, or black scales. Forewings with fork II are present, M bifurcated once, the discoidal cell is not large, and thyridial cell is short and in a basal position, Cu_1_ simple, and two or three Anal veins are present. Hind wings with fork V can be present or absent, Sc is usually absent, R_1_ is long and ending at wing margin. R_2 + 3_ is absent, cross-veins *r* and *r m* are long, with the bifurcation of Rs before that of M.

Male genitalia: Tergite VIII overlaps base of segment IX and its acrotergite. Sternite IX has a pair of anterior lateral apodemes. Its apicoventral margin developed into a large tongue. Preanal appendages are enlarged at their tips. Intermediate appendages are present, weakly sclerotized. Segment X is cylindrical. Inferior appendages with two segments are fused and discernable, each with spines on inner sides at their bases. The aedeagus is membranous and long.

The species groups according to Schmid (1982) are defined by the following characteristics:

*Drepanocentron druhyu* species group: Each of the forewings has two anal veins. Hind wings, R_1_ and Rs, are distinct at the base, early bifurcation of Rs, transverse R_1_-Rs closer to Rs-M level. Inferior appendages second article is with a rounded lobe at the base. The aedeagus has two parallel sclerites.

*Drepanocentron citrangada* species group: Each of the forewings usually has three anal veins. Hind wings, R_1_ and Rs, are united basally, late bifurcation of Rs, transverse R_1_-Rs far from Rs-M level, forming a false discoidal cell. Inferior appendages’ lower margins are straight, while the lower basal part of the second article does not develop into a lobe. The aedeagus tip has a simple, odd lobe.

### 3.1.3. Key to Three Genera of Xiphocentroninae

1.Species in Oriental and Afrotropical regions with inferior appendages not bifurcated in lateral view           2-Species in Oriental region with inferior appendages bifurcated in lateral view or species in Neotropical region    other genera of Xiphocentroninae2.Forewings dark brown, with several white spots or stripes             *Abaria*-Forewings unicolorous, without obvious color spots   33.Forewing fork IV present, inferior appendages, second article inner side with comb-like process or row of bristles.       *Melanotrichia*-Forewing fork IV absent, inferior appendages, second article inner side with scattered spines.         *Drepanocentron*

### 3.2. Description

#### 3.2.1. *Abaria herringbona* n. sp.

Description: Specimens were in alcohol, with compound eyes black, wings pale brown, legs yellowish brown. Fore- and hind wings are with patches of scaloid setae. Each of the forewings (Figure 1a) has a length of 2.5–2.6 mm (*n* = 2); venation is typical of the genus, but Sc is extended beyond the bifurcation of Rs, fork II sessile. Hind wings are 2.1–2.2 mm long (*n* = 2), the venation is typical of the genus, but crossvein *r m* originating at the bifurcation of fork II. Spurs: 0-4-3.

Male genitalia: Tergum VIII shown in lateral view is sub-rectangular (Figure 1b), with apicoventral angles produced ventrad; in dorsal view, apical margin convex and anterior margin incised mesally (Figure 1d). Segment IX tergum is almost indiscernible, retracted under tergum VIII in lateral view (Figure 1b); sternum ovoid is longer than high in lateral view, with a pair of anterior apodemes that are long and slender (Figure 1b). Segment X is tongue shaped in lateral view and triangular in dorsal view, with apical tip narrowly incised mesally 2/3 its length (Figure 1b,d). Preanal appendages in lateral view are elongated and clavate, each with base narrow, distal 3/4 curved, and setose (Figure 1d); in dorsal view, each is observed with a middle portion of inner margin bulging inward (Figure 1d). Inferior appendages’ first and second articles are completely fused, each in lateral view tapering from base to pointed apex, while in ventral view, basal half is directed laterad then curved caudad with distal half almost straight, with subapical spines on the inner side. Phallic apparatus is long and tube-like, with base bulging and apex dilated (Figure 1e).

Diagnosis: The new species belongs to Schmid’s [4] *A. tripunctata* species group since its preanal appendages are slender and cylindrical, and their lower margins do not form a blunt angle. It resembles *A. dunga* Oláh & Johanson 2010 from Vietnam in the shapes of preanal and inferior appendages, but it can still be distinguished by the following features: (1) forewing Sc is long, extending beyond the bifurcation of Rs; Cu_1_ and Cu_2_ are separated from each other widely, not fused apically; (2) segment VIII in dorsal view with apical margin is slightly convex and anterior margin incised; (3) the distal end of segment X is strongly incised and base sixth is wider in dorsal view.

Holotype: Male, P.R. China, Guang-xi Province: He-chi City, Jin-cheng-jiang District, Liu-jia-he Town, 24.7314°N, 107.8956°E, alt. 165 m, 10 May 2020, Malaise trap, coll. X. Lin (NJAU).

Paratypes: one male, same data as holotype; three males, P.R. China, Guang-xi Province: He-chi City, Jin-cheng-jiang District, Liu-jia-he Town, 24.7316°N, 107.8961°E, alt. 150 m, from July 20 to 2 August 2020, Malaise trap, coll. X. Lin (NJAU).

Distribution: China (Guang-xi).

Etymology: *herringbona*, name derived from English-language noun *herringbone, -a*, refers to the shape of the anterior margin of male tergum VIII in dorsal view.

#### 3.2.2. *Drepanocentron fuxiensis* n. sp.

Description: Forewing length is 3.2 mm. The body color is light yellow (in alcohol); head and wings are brown, with compound eyes black. Forewings are pale brown, and the venation is typical of the genus, with apical fork II and two anal veins. Hind wings with fork V are present, and a false vein between R and M is obvious (Figure 2a). Spurs 2-4-3; apical spur of each hind leg (Figure 2f) is modified into a lance-shaped process.

Male genitalia: Tergum VIII is roof shaped in lateral view, fused with tergum IX, while in dorsal view, its anterior margin is deeply concave and its posterior margin shallowly concave. Tergum IX in lateral view is tongue shaped but in dorsal view is subrectangular, with apical margin shallowly concave. Sternum IX in lateral view is an elongated oval, with anterior margins produced into lateral apodemes (Figure 2d); in ventral view, the posterior margin middle plate of sternum IX is shallowly notched (Figure 2c). Segment X is weakly sclerotized and subrectangular in lateral view (Figure 2d), with each posteroventral angle produced posterad; in ventral view, apex has deep incision (Figure 2c). Preanal appendages are elongated and clavate in lateral view (Figure 2d); in dorsal view, distal half is wider than basal half, with setose. Inferior appendages’ first and second articles are completely fused and in lateral view are elongated triangular; in ventral view, one-third of the basal is broad and the other two-thirds taper to rounded apex, with inner side scattered with spines. Phallic apparatus is tube-like, base enlarged, with subapical portion slightly swollen (Figure 2b,d,e).

Diagnosis: The new species belongs to Schmid’s [4] *Drepanocentron druhyu* species group with the forewings having two anal veins, and somewhat resembles *D. jiska* Malicky 2009 from Vietnam in the inferior appendages when viewed ventrally. It can be distinguished from the latter by the following features: (1) the apical spurs of the hindlegs, each with the ratio of length to width at 9:2, rather than 9:1 as in *D. druhyu*; (2) in each of the preanal appendages, the distal half is broader than the basal one, rather than nearly of the same width as in *D. druhyu* in dorsal view; (3) the mesal plate of sternum IX is much more slender than that of *D. druhyu*; (4) the upper edges of the inferior appendages are straight in lateral view, with lower margin sinuate, rather than straight, as in the latter species.

Holotype: Male, P.R. China, An-hui Province: Huang-shan City, Huang-shan District, Tang-kou Town, Fu-xi Village, Fu-xi River, 30.0698°N, 118.1589°E, alt. 450 m, 12 July 2021, light trap, coll. C. Sun & L. Peng (NJAU).

Distribution: China (An-hui).

Etymology: Fuxiensis, name derived from the type locality.

#### 3.2.3. Melanotrichia attia Malicky & Chantaramongko 1992

*Melanotrichia attia* Malicky & Chantaramongkol 1992: 20.

Type locality: Thailand, Doi Inthanon, 1600 m, 25 March 1992.

This species was originally described in Doi Inthanon National Park of Thailand by Malicky and Chantaramongkol [15].

New records: one male, P.R. China, Gui-zhou Province: Qian-dong-nan Miao and Dong Autonomous, Fang-xiang town, Ge-tou Village, Lei-gong-shan Natural Reserve, 26.3960° N, 108.2609° E, alt. 1070 m, from 10 to 20 April 2021, Malaise trap, coll. X. Lin; 1 male, P.R. China, Gui-zhou Province: Qian-dong-nan Miao and Dong Autonomous, Fang-xiang town, Ge-tou Village, Lei-gong-shan Natural Reserve, 26.3960° N, 108.2609° E, alt. 1070m, from 20 June to 6 July 2021, Malaise trap, coll. X. Lin (NJAU).

#### 3.2.4. *Melanotrichia acclivopennis* (Hwang, 1957)

Kibuneopsychomyia acclivopennis Hwang, 1957, Schmid, 1982: 3

Type locality: China, Fu-Jian Province, 22 May 1945.

This species was originally collected in Wu-yi-shan National Park of China by C. Fu [33].

Materials examined: one male, P.R. China, Zhe-jiang Province: Leng-keng, Hang-he Town, An-ji County, Hu-zhou City, 30.51°N, 119.37°E, alt. 421 m, 8 May 2015, light trap, coll. Y. Hu (NJAU).

Distribution: China (Fu-jian, Guang-dong, Zhe-jiang)

## 4. Discussion

Xiphocentronidae species have fairly constant ecological characteristics occurring most abundantly in the tropical regions. There are 118 species distributed in the Oriental region and 59 species in the Neotropical region. The two together account for more than 95% of the world’s total. The genera *Drepanocentron* and *Abaria* are present on islands of Indonesia, Borneo, and the Philippines, with *Abaria* occurring even in New Guinea. However, the genus *Melanotrichia*, which is diverse in continental Asia, has not been found on these islands. This suggests that *Drepanocentron* and *Abaria* were present in Southeast Asia before *Melanotrichia*, and thus, they could have dispersed between these islands during the Pleistocene glacial periods, when the sea level was low, and there was connectivity from Borneo, Java, and Sumatra to the Malay Peninsula and mainland Asia [32,34].

The existing records [8,11] showed that China has recognized only three species in the genus *Melanotrichia*, which are distributed in the Oriental region. The spread of *Melanotrichia* on the Asian continent may have been hindered by the topography of the Qin-ling Mountains and Huai-he River. This could be a reason why a total of three species of the East Palearctic region in the *Melanotrichia* genus, all distributed in Japan, have no record in the vast Chinese East Palearctic area. In addition, the collection efforts in the northern part of China are not as good as those in the south, which may have limited our knowledge of the actual distribution of Xiphocentronidae.

Thus far, China has recorded six species of Xiphocentronidae in three genera, all distributed in the Oriental region (Figure 3). They are often more active during the day when the temperature is higher [6]; only two of the nine xiphocentronid specimens we collected with light traps, which also reflects this observation well. In order to increase the known diversity of the family Xiphocentronidae in China, we need to increase the proportion of sweep netting and Malaise trapping in the collection operations. There is a vast area rich in streams that has a humid climate and suitable water temperatures in southern China. Considering that there are good natural conditions suitable for their growth, it is reasonable to expect that xiphocentronids have a higher diversity in China than currently known.

## 5. Conclusions

Many Trichoptera are without a strong ability to fly; the majority of their life cycle is spent as larvae in the water. The spread and differentiation of their species are mostly resulting from the spread and change of the watershed. The Discovery of *Drepanocentron fuxiensis* sp. nov. and *Abaria herringbona* sp. nov. extends the range of the two genera to the northeastern part of the Oriental region. These two species records will provide distribution data to phylogeographic analyses and increasing knowledge of the biodiversity and species distribution may help in further studies.

## Figures and Tables

**Figure 1 insects-13-00095-f001:**
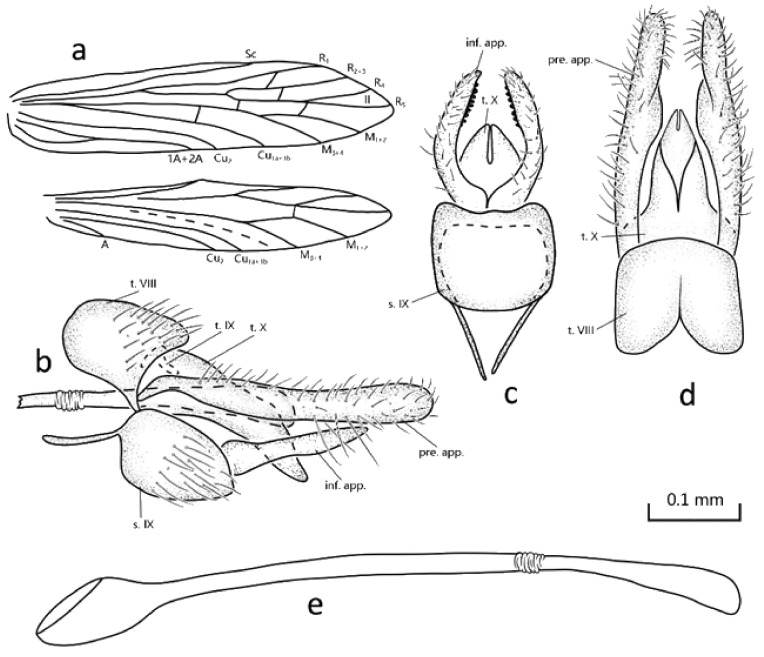
Male genitalia and wing venation of *Abaria herringbona* n. sp. and Supplementary Information: (**a**) wing venation; (**b**) left lateral view; (**c**) ventral view; (**d**) dorsal view; (**e**) phallic organ, left lateral view. Abbreviations: inf. app., inferior appendages (paired); pre. app., preanal appendages (paired); s. IX, sternum IX; t. VIII, tergum VIII; t. IX, tergum IX; t. X, tergum X. Scale bar refers to (**b**–**e**).

**Figure 2 insects-13-00095-f002:**
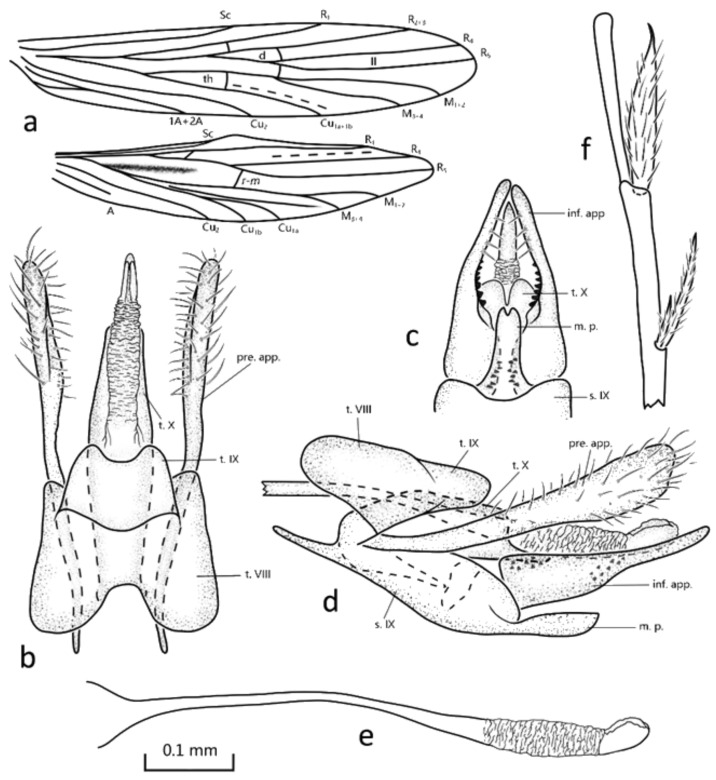
Male genitalia and wing venation of *Drepanocentron fuxiensis* n. sp: (**a**) wing venation; (**b**) dorsal view; (**c**) ventral view; (**d**) left lateral view; (**e**) phallic organ, left lateral view; (**f**) modified spur on left hind leg. Abbreviations: inf. app., inferior appendages (paired); m. p., middle plate of sternum IX; pre. app., preanal appendages (paired); s. IX, sternum IX; t. VIII, tergum VIII; t. IX, tergum IX; X, tergum X. Scale bar refers to all Figure 2b–e.

**Figure 3 insects-13-00095-f003:**
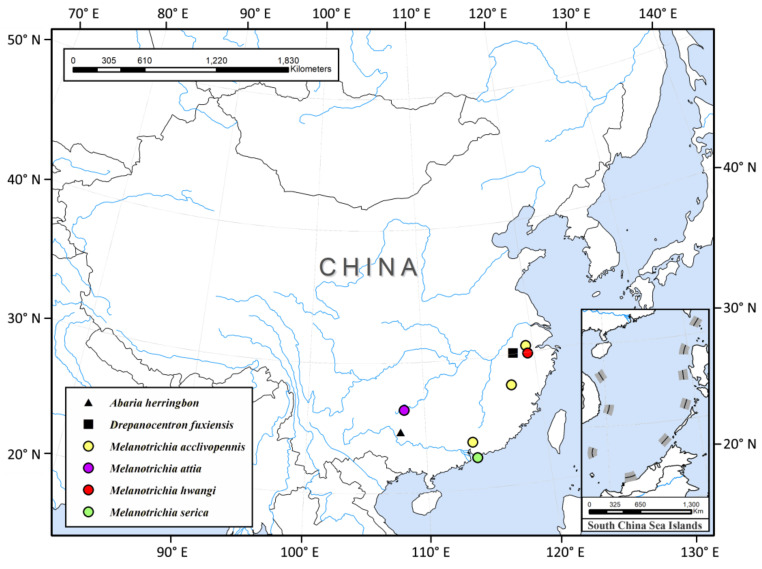
Distribution of Xiphocentronidae species in China.

## Data Availability

All data are available in this paper.

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
