# Peer review of "The First Report of the Genera Abaria and Drepanocentron (Trichoptera: Xiphocentronidae) from China, with Descriptions of Two New Species"

_insects, 2022, doi:10.3390/insects13010095_

Round 1

Reviewer 1 Report

This manuscript summarizes knowledge about the family Xiphocentronidae in China, describes two species that are new for science in two genera of the family not previously known in China. Furthermore, the manuscript provides new records for China of a species that previously was known only from Thailand.

In general, the scientific content of the manuscript is consistent with the best modern professional standards. The review of current knowledge of the global and Chinese fauna for the subject family and genera in the Introduction is complete and accurate with appropriate bibliographic references. Both the descriptions and diagnoses of the new species and the newly recorded species are very good, allowing future identification of these species and study of their other life forms and their biology.

However, the manuscript is not publishable in its present form because it is in need of significant revision of the English language, with important changes needed in most sentences.

Author Response

Dear reviewer:

Thank you very much for your suggestions.

Point 1: “However, the manuscript is not publishable in its present form because it is in need of significant revision of the English language, with important changes needed in most sentences.”

Response 1: As you suggested, the manuscript has been edited by MDPI English editor, and we checked the changes after editing the English to make sure that haven't changed our original intention.

Other main amendments are as follows:

1.Introduction

  • The number of known species in the genus Drepanocentron has been corrected from 41 to 42. The newly added species are distributed in Vietnam, and the corresponding distribution and records have been changed.
  • The number of recorded Xiphocentronidae species in China has been corrected from 2 to 3. Melanotrichia serica Barnard & Dudgeon, 1984) has been added.
  1. Materials and Methods
  • One species from Zhe-jiang by light trap as Melanotrichia acclivopennis (Hwang, 1957) has been added.
  • The terminology for setal warts is from Olah & Johanson and corresponding literature has been added.

  1. Results
  • The restatement language of the defining characters of the genera and species groups has been modified and supplemented.
  • Key to 3 genera (which with records of distribution in China) of Xiphocentroninae has been added.
  • The name of the new species "Abaria herringbon" was revised to "Abaria herringbone".
  • Added the record of Melanotrichia acclivopennis in the new province in China.

  1. Discussion
  • Rewritten the discussion section and added a distribution map with the location of all the Xiphocentronidae species recorded in China.

Best regards,

Lang Peng

Reviewer 2 Report

It is a long overdue discovery  of the genera Drepanocentron and Abaria in China. I do not have much criticism. It is a well written taxonomic paper. However. in the chapter Methods, some sentences about the collecting methods are necessary. In the chapter Results, the subchapter Taxonomy is actually not important and could be deleted. But it can be retained, if the subchapter ends with an identification key for all 3 genera.

In the acknowledgements the persons should be mentioned, who were helpful in correcting the English text and in checking the taxonomic correctness.

Author Response

Dear reviewer:

Thank you very much for your suggestions.

Point 1: “However. in the chapter Methods, some sentences about the collecting methods are necessary. In the chapter Results, the subchapter Taxonomy is actually not important and could be deleted. But it can be retained, if the subchapter ends with an identification key for all 3 genera.”

Response 1: We would like to save the subchapter Taxonomy. As you suggested we added the identification key for all 3 genera (which with records of distribution in China) of Xiphocentroninae.

Point 2: “In the acknowledgements the persons should be mentioned, who were helpful in correcting the English text and in checking the taxonomic correctness.”

Response 2: The manuscript has been edited by MDPI English editor, and we checked the changes after editing the English to make sure that haven't changed our original intention. We added reviewers, editor and English editor to the acknowledgment, as it should be.

Best regards,

Lang Peng

Reviewer 3 Report

*I included in the PDF some comments, suggestions, corrections. The aim is to make it concise, clear.

1) The results (the proposition of 2 new species and the new records in China) are relevant.

2) The illustrations are good and clear.

3) The general description/diagnosis of the genera Drepanocentron and Abaria were translations of the Schmid’s (1982) original descriptions in french. The restatement of the defining characters of the genera and species groups can help other people to work with the group. Although, the original description as it appears in french has imprecise terms and inadequate style for the modern taxonomic standards. So, you should fix this issues when presenting the characters, otherwise would be better just reference to Schmid’s original study and present your new species hypothesis.

To do it so, you should write all the descriptions in the taxonomy section in telegraphic style; the character/structure should be presented first and followed by the character states/adjetives; and imprecise and less informative words should be evoided.

The following paper is a good guide:

https://www.researchgate.net/publication/269581236_Best_Writing_and_Curatorial_Practices_for_Describing_a_New_Species_of_Beetle_a_Primer

4) The introduction contains text identical to Holzenthal et al. (2007), also this paper was not referenced. The introduction should be rewrite. Holzenthal et al. (2007) paper: https://www.mapress.com/zootaxa/2007f/zt01668p698.pdf

5) Parts of the text have wrong spelling and are not smooth to reading. I did not indicate all of then in the PDF, so consider sending the text to be checked by a native speaker.

6) I suggest the inclusion of a distribution map with the location of all the Xiphocentronidae species recorded in China. This distribution placed over the major watershed and rivers could help in the discussion of potential areas of biodiversity to be explored and the relationship between regional fauna and species.

7)The discussion as it was presented, did not talk about the paper results. I suggest that the discussion be about the new species, their possible biogeographical and evolutionary significance and the contribution of the study to the knowledge of Chinese caddisfly diversity. 

These papers on freshwater bivalves may help to associate the organism distribution with the history of the East Asian drainage: https://www.nature.com/articles/s41598-020-63612-5.pdf
https://www.researchgate.net/publication/235354901

Best regards!

Round 2

Reviewer 1 Report

Please see recommendations to improve English grammar and composition marked in the text.

Reviewer 3 Report

The paper was significantly improved but still have language issues on the newly added parts. The distribution map is a good improvement for the paper.
A couple of references of the species numbers on the introduction need to be updated, also the number of genera in the family changed to eight in a paper that was just published.
In the results, I suggest the exclusion of the generic description of the family, since the paper is not a revision. Many characters are so variable in the family that the general description turns out completely uninformative. Also, Abaria and Drepanocentron as described by Schmid (1982), usually have relevant characters in the head and thorax. Ex.: Abaria do not have mesoprescutum, and Drepanocentron a somewhat rounded one. Abaria often has distinct setae on antennae, and legs; the mesoscutellum also can be of different shapes in the species. The head coronal and occipital sutures also vary sometimes in Abaria and Drepanocentron.
So, the informative description of each holotype head and thorax would be more appropriate. Rather than coping an ordinary description of head and thorax from Olah & Johanson (2010), I suggest to just look at the holotypes of the two genera and do a simple description of the structures that you see that are different between them. If the structures are identical in two different genera they probably will not be relevant at all and you may skip them.
Other corrections and suggestions are given in the PDF.

Best regards.
